# mFFE CT-like MRI Sequences for the Assessment of Vertebral Fractures

**DOI:** 10.3390/diagnostics14212434

**Published:** 2024-10-30

**Authors:** David Ferreira Branco, Hicham Bouredoucen, Marion Hamard, Karel Gorican, Pierre-Alexandre Poletti, Bénédicte Marie Anne Delattre, Sana Boudabbous

**Affiliations:** Geneva University Hospitals, Rue Gabrielle-Perret-Gentil 4, 1205 Geneva, Switzerland; david.ferreirabranco@hcuge.ch (D.F.B.); marion.hamard@hcuge.ch (M.H.); karel.gorican@umontreal.ca (K.G.); pierre-alexandre.poletti@hcuge.ch (P.-A.P.); benedicte.delattre@hug.ch (B.M.A.D.); sana.boudabbous@hcuge.ch (S.B.)

**Keywords:** magnetic resonance imaging, CT-like, bone, disc and ligamentous lesions, spine, spinal fracture

## Abstract

Objectives: The aim of this study was to evaluate the diagnostic performance, image quality, and inter- and intra-observer agreement of the 3D T1 multi-echo fast field echo (mFFE) sequence in cervico-thoraco-lumbar vertebral fractures compared with conventional computed tomography (CT) as the gold standard. Methods: We conducted a prospective single-centre study including 29 patients who underwent spinal magnetic resonance imaging (MRI) at the surgeon’s request, in addition to CT for vertebral fracture assessment and classification. A 3D T1 mFFE sequence was added to the standard MRI protocol. Consecutively, two readers analyzed the 3D mFFE sequence alone, the 3D mFFE sequence with the entire MRI protocol, including the STIR and T1 sequences, and, finally, the CT images in random order and 1 month apart. A standardized assessment was performed to determine the presence or absence of a fracture, its location, its classification according to the Genant and AO classifications for traumatic and osteoporotic fractures, respectively, the loss of height of the anterior and posterior walls of the vertebral body, and the presence of concomitant disco-ligamentous lesions. Contingency tables, intraclass correlation coefficients, and Cohen’s kappa tests were used for statistical analysis. Results: A total of 25 fractures were recorded (48% cervical, 20% thoracic, and 32% lumbar), of which 52% were classified A, according to the AO classification system. The quality of the 3D mFFE image was good or excellent in 72% of cases. Inter-observer agreement was near perfect (0.81–1) for vertebral body height and for AO and Genant classifications for all modalities. Intra-observer agreement was strong-to-near perfect between CT and the 3D mFFE sequence. Regarding the diagnostic performance of the 3D mFFE sequence, the sensitivity was 0.9200 and 0.9600, the specificity was 0.9843 and 0.9895, and the accuracy was 0.9861 and 0.9769 for Readers 1 and 2, respectively. In addition, up to 40% of intervertebral disc lesions and 33% of ligamentous lesions were detected by the 3D mFFE sequence compared to CT, allowing four AO type A fractures to be reclassified as type B. Conclusions: The 3D mFFE sequence allows accurate diagnosis of vertebral fractures, with superiority over CT in detecting disco-ligamentous lesions and a more precise classification of fractures, which can prompt clinicians to adapt their management despite an image quality that still requires improvement in some cases. Key points: Vertebral fractures and disco-ligamentous lesions can be assessed using CT-like MRI sequences, with 3D T1 mFFE being superior to CT for the detection of disco-ligamentous lesions. CT-like images using the 3D T1 mFFE sequence improve the diagnostic accuracy of bone structures in MRI.

## 1. Introduction

Vertebral fractures are associated with back pain and disability that worsen with the number and severity of the fractures [1,2,3] and with increased mortality, regardless of gender [4,5]. Vertebral fractures are best assessed by analyzing the bony structures in addition to the medullary, discal, and ligamentous structures, as well as the perivertebral soft tissues, using complementary computed tomography (CT) and magnetic resonance imaging (MRI) modalities. Despite being essential for the assessment of vertebral fractures, the combined use of these two imaging modalities adds significant costs to the healthcare system, delays diagnostic management, and exposes patients to radiation from the use of CT [6,7]. With this in mind, a number of studies have proposed the use of new high-resolution MRI sequences to complement the usual MRI protocol for exhaustive bone assessment, with the aim of replacing CT in the evaluation of vertebral fractures, degenerative changes [8,9,10], or bone tumours [11]. Using MRI-derived CT-like images based on ultra-short echo time (UTE) [8,9], high-resolution 3D T1-weighted spoiled gradient echo [8,9,10,11] (T1SGRE), or susceptibility-weighted imaging (SWI) [10] sequences, these studies have demonstrated substantial-to-perfect diagnostic performance, inter-observer agreement, and the morphological assessment of fractures compared with conventional CT. The 3D T1 multi-echo fast field echo (mFFE) sequence is a high-resolution 3D gradient-echo technique that uses multiple echoes with constant spacing, corresponding to in-phase time echoes based on an MRI scanner field strength of 4.6 ms at 1.5 T. This provides superior cortical and trabecular bone contrast and contours, minimizing chemical shift for better delineation and localization of the bone with reduced edge blurring, and reducing the additional dephasing caused by T2* decay to help reduce signal loss at bone tissue interfaces. After acquisition, two additional post-processing steps are performed to produce images with contrast closer to that of CT. The first post-processing step consists of summing the amplitude of all the echoes, which increases the signal-to-noise ratio in the sequence. After summation, the images from the last echo are subtracted from the summed images to invert the grayscale and provide the bone with a CT-like contrast [12].

The aim of this study was to evaluate the inter- and intra-observer agreement, diagnostic performance, and diagnostic quality of the 3D T1 mFFE sequence in the morphological assessment of cervico-thoraco-lumbar vertebral fractures compared with conventional CT as the gold standard.

## 2. Material and Methods

### 2.1. Patient Population

In this prospective single-centre study, all consecutive patients admitted to the emergency department of our institution between August 2021 and November 2021 were screened for possible participation in the study. Inclusion criteria were suspected acute cervical, thoracic, or lumbar vertebral fractures for which a CT scan was performed as part of the diagnostic process, with an additional MRI carried out within 48 h of the trauma if concomitant ligamentous injury was suspected.

Of the twenty-nine patients identified as eligible for the study, nine were excluded for various reasons: four patients underwent surgery without MRI, three had a contraindication to MRI, and two refused to undergo MRI (see Figure 1).

The protocol was approved by the institutional ethics committee, Geneva University Hospitals (CCR number: 2017–01276).

### 2.2. Imaging Protocol and Analysis

CT images were acquired on a Somatom Force CT scanner (Siemens Healthineers, Forchheim, Germany) with the following parameters: pixel spacing, 0.2948/0.2949; pitch factor, 0,8; collimation, 0.6 mm; tube voltage peak, 120 kV; modulated tube current, 250 mA. Images reformatted in 1.5 mm slice thickness were analyzed in a bone kernel (window width 1500; window level 300). MRI images were acquired on a 1.5-T Philips Ingenia scanner (Philips Healthcare, Best, The Netherlands) with a head–neck 20-channel and flex-coverage osterior 44-channel coils. In addition to the usual MRI protocol (T1 spin echo sagittal; T2 spin echo sagittal; T2 STIR [short TI inversion recovery] sagittal; T2 spin echo axial) performed in the context of a traumatic spine, a 3D mFFE sequence (with 4 multiple echoes) was also implemented (see Table 1). CT-like images were reconstructed according to Johnson B et al. [12]. Images were analyzed using Osirix MD (Pixmeo SARL, Bernex, Switzerland) under anonymized conditions. Two radiologists specializing in osteoarticular disorders with 10 years (M.H) and 7 years (K.G) experience, respectively, performed the measurements separately, with an interval of approximately 4 weeks between readings of the different imaging modalities, including CT images alone, CT-like images alone, and the combination of CT-like images and other MRI sequences from the routine spine protocol. To minimize observer variability, we applied a standardized protocol to record the presence or absence of a fracture, its location, and morphological data, including the loss of vertebral height according to the Genant classification, loss of height of the anterior and posterior walls of the vertebral body, fracture classification according to the AO classification, the presence of concomitant disco-ligamentous damage, and diagnostic quality of the images using a Likert scale (see Table 2). For one case, a consensual lecture was performed to avoid the inter-individual variability of measurements.

### 2.3. Statistical Analysis

To estimate inter- and intra-observer agreement, intraclass correlation coefficients (ICCs) were used for numerical, normally distributed data, and weighted Cohen’s kappa was used for ordinal data, with 95% confidence intervals (CIs) calculated for each value. To interpret the agreement statistic scores, we used the criteria developed by Landis and Koch [13]. According to these criteria, values from 0 to 0.20 represent poor agreement, 0.21 to 0.40 moderate agreement, 0.41 to 0.60 moderate agreement, 0.61 to 0.80 strong agreement, and 0.81 to 1 almost perfect agreement.

Contingency tables were used to assess the diagnostic performance of CT-like imaging in detecting vertebral fractures. MDCalc software version 23.0.6 (Ostend, Belgium) was used for these statistical analyses.

## 3. Results

Overall, 20 patients were examined in this study, with a mean age of 58.3 years and a sex ratio of 0.65 in favour of males. A total of 25 fractures were recorded on CT, yielding an average of 1.25 fractures per patient (range 1–4). Of these, 48% were cervical fractures (*n* = 12), 20% were thoracic fractures (*n* = 5), and 32% were lumbar fractures (*n* = 8). According to the AO classification, 52% of the fractures were classified A (*n* = 13), 44% were classified B (*n* = 11), and 4% were classified C (*n* = 1) (see Figure 2a,b).

Using a Likert scale, the observers rated the diagnostic quality of the images with a median score of four (good). Specifically, 4% (*n* = 1) of cases were rated inadequate, 20% (*n* = 5) were rated poor, 4% (*n* = 1) were rated moderate, 52% (*n* = 13) were rated good, and 20% (*n* = 5) were rated excellent.

Inter-observer agreement was near perfect (0.81–1) for all quantitative (vertebral body height) and ordinal (AO and Genant classifications) parameters and for both modalities. Inter-observer agreement was slightly higher for all CT measurements than CT-like measurements (see Table 3). Given the high inter-observer agreement, the intra-observer agreement was calculated based on the measurements of the most experienced observer (M.H). The intra-observer agreement between CT and CT-like images based on the measurements of Observer 1 was strong (0.61–0.80) for the Genant classification (0.79039; 95%CI 0.60741–0.97338) and near perfect (0.81–1) for the AO classification (0.87027; 95%CI 0.73079–1.000), the anterior vertebral body height (0.9600; 95%CI 0.9098–0.9825), and the posterior vertebral body height (0.9757; 95%CI 0.9447–0.9894). 

Using CT as the gold standard, we calculated the diagnostic performance of the mFFe CT-like MRI sequence. Observer 1 detected 24 of the 25 fractures visualized on CT with two false positives and one false negative, resulting in a sensitivity of 0.9600, a specificity of 0.9895, and an accuracy of 0.9861. Observer 2 detected 23 of the 25 fractures visualized on CT, with two false positives and three false negatives, resulting in a sensitivity of 0.9200, a specificity of 0.9843, and an accuracy of 0.9769. 

Regarding the detection of ligamentous and intervertebral disc lesions, reading the CT-like images alone versus reading the CT images alone allowed for the detection of 20% (*n* = 1) of intervertebral disc lesions and 33% (*n* = 4) of ligamentous lesions for Observer 1, while it allowed the detection of 40% (*n* = 2) of intervertebral disc lesions and 25% (*n* = 3) of ligamentous lesions for Observer 2, out of a total of five intervertebral disc lesions and twelve ligamentous lesions, which was confirmed by reading the CT-like images along with the complete spinal MRI protocol. Reading the CT-like images resulted in the reclassification of four fractures originally classified as AO type A on CT to type B in each observer.

## 4. Discussion

The purpose of this study was to evaluate the diagnostic performance, as well as the inter- and intra-observer agreement of CT-like sequences. Inter-observer agreement for CT-like sequences and intra-observer agreement between CT and CT-like images for morphological assessment were in line with the literature [8,9,10,11], with weighted kappa and ICC greater than or equal to 0.90 for quantitative and ordinal parameters. Inter- and intra-observer agreement were better for quantitative parameters (anterior and posterior vertebral body height [ICC > 0.95]) than for ordinal parameters (AO and Genant classifications [weighted kappa > 0.90]), which is consistent with data in the literature [8,9,10].

The diagnostic performance of the CT-like sequence in detecting fractures was remarkable for both observers, with respective sensitivity, specificity, and accuracy of 0.9600, 0.9895, and 0.9861 for Observer 1, and 0.9200, 0.9843, and 0.9769 for Observer 2, which is in accordance with the previously published results by Schwaiger et al. [8] (sensitivity 0.95/0.93; specificity 0.98/0.98; and accuracy 0.97/0.97 for both observers [8]). We attribute these slight differences to the diagnostic performance between the two observers and the difference in experience between them. These results are very promising and highlight the ability of these sequences to reach the level of CT in the bone assessment of vertebral fractures without significantly increasing the management time, extending the duration of the MRI from a minimum of 3 min 46 s for a cervical acquisition to a maximum of 14 min 4 s for a cervico-thoraco-lumbar acquisition, with the total MRI protocol time being at least 13.5 min for a cervical MRI (see Table 1). However, this sequence cannot completely replace CT in trauma cases, especially when extraosseous visceral lesions are suspected; in these cases, the combination of CT and MRI remains essential despite a longer acquisition time of at least 26.5 min (13 min for the polytrauma CT whole-body protocol and at least 13.5 min if MRI is performed only at the cervical level).

Studies have demonstrated the value of specific CT-like sequences in the assessment of soft tissues, notably the ligament, muscle, and tendon structures [14,15,16], in particular 3D gradient-echo sequences, including the 3D T1 mFFE sequence we used. Our study showed that reading CT-like sequences alone led to the detection of four ligamentous lesions in Observer 1 that were not visualized when reading the CT images alone and three ligamentous lesions in Observer 2, representing 33% of all ligamentous lesions confirmed when reading the CT-like sequences together with the rest of the MRI protocol (*n* = 12). Our results confirm the data in the literature and are perfectly illustrated in the following images (see Figure 3).

The improved contrast [14,15,16] within the soft tissues compared to CT may also be of interest in detecting the edematous infiltration [16] of these tissues, as illustrated in Figure 4.

Gas accumulations in the intervertebral discs are devoid of signal and appear bright on the inverted reformatted CT-like images [8]. The two cases (see the Results) in which disc lesions were detected by reading the CT-like sequences alone, which were not visible on CT and were confirmed on MRI, correspond to cases in which the readers visualized intra-discal air that is not specific to a traumatic origin [17], and therefore, cannot be used as the sole diagnostic criterion for a traumatic disc lesion.

The evaluation of all these parameters allowed 16% (*n* = 4/25) of fractures to be initially classified as A, according to the AO classification when reading CT images alone, to be reclassified as B after reading the CT-like images prior to their correlation with the rest of the MRI protocol. The use of this sequence may, therefore, play a role in the classification of vertebral fractures [18] and, consequently, modify therapeutic management [19,20].

The median score for the diagnostic quality of these images was four (good) on the Likert scale, which is slightly lower than the results in the literature [8].

Our study has several limitations. Although the distribution of fractures in our population is representative of all cervical, thoracic, and lumbar levels, with fractures defined as stable and unstable classified as A, B, and C according to the AO classification, including translation and distraction lesions, our population was relatively small, and similar studies need to be performed in larger populations. 

No pathological fractures with bone lesions were included in this study, and the value of such a sequence as an adjunct to MRI for this type of fracture remains to be demonstrated.

Despite the use of 1.5-T MRI, no patients with spondylodesis material were included in this study. The presence of metallic material and the potentially associated artefacts must be considered, and the CT-like sequence must be evaluated in this situation to determine its usefulness.

Spectral CTs encompass several CT techniques, including dual-energy (DECT) and, more recently, photon-counting CT. There is a new interest in this imaging technique because of its physical principle, which is based on the simultaneous acquisition of data at multiple energy levels, allowing for the better characterization and differentiation of the structures under study [21].

Recent studies have focused on this technique to assess the presence of bone marrow edema, using virtual non-calcium (VNCa) reconstruction algorithms that suppress the high attenuation of trabecular bone by reducing or eliminating the presence of calcium, allowing for better visualization of the underlying marrow for the detection and characterization of vertebral fractures [22]. These studies [23,24] have demonstrated sensitivities of over 80% and specificities of almost 100%, highlighting the value of this modality, particularly for assessing the acute nature of a fracture. However, the assessment of ligaments is essential for the examination of unstable fractures, and, in these cases, the use of the CT-like sequence instead of CT or DECT could allow the detection of lesions before confirmation by the rest of the MRI protocol. We could imagine using this sequence as a screening technique that may or may not lead to a full MRI protocol. However, further studies would be needed to confirm these hypotheses.

Despite the potential of CT-like sequences, it is important to bear in mind that their use requires the widespread availability of MRI, which is not the case in many parts of the world.

## 5. Conclusions

Three-dimensional T1 mFFE CT-like sequences have a high diagnostic performance and have shown strong-to-perfect agreement with CT for the detection of vertebral fractures while allowing the detection of ligamentous lesions in potentially unstable fractures, leading to a full MRI protocol in these patients.

## Figures and Tables

**Figure 1 diagnostics-14-02434-f001:**
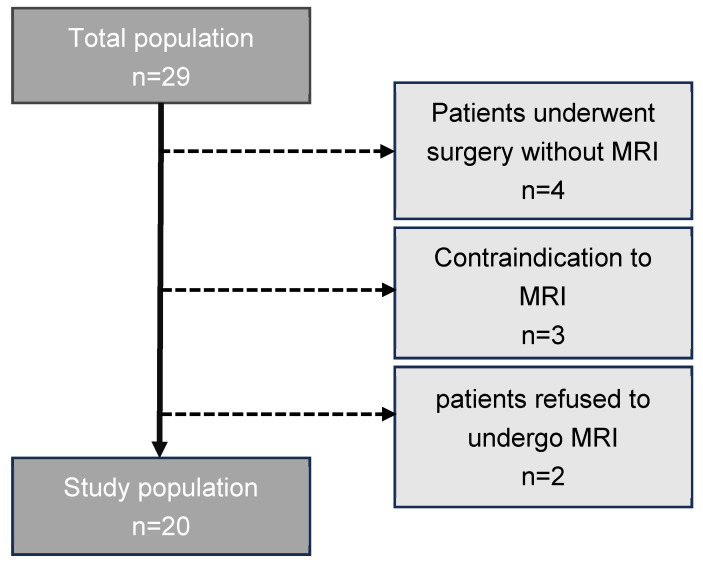
Patient population.

**Figure 2 diagnostics-14-02434-f002:**
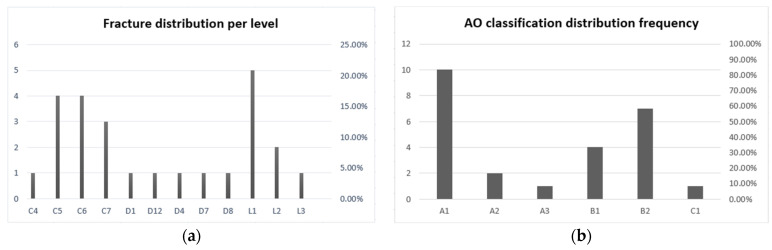
(**a**) Distribution of fractures per vertebral level; (**b**) distribution of fractures according to the AO classification.

**Figure 3 diagnostics-14-02434-f003:**
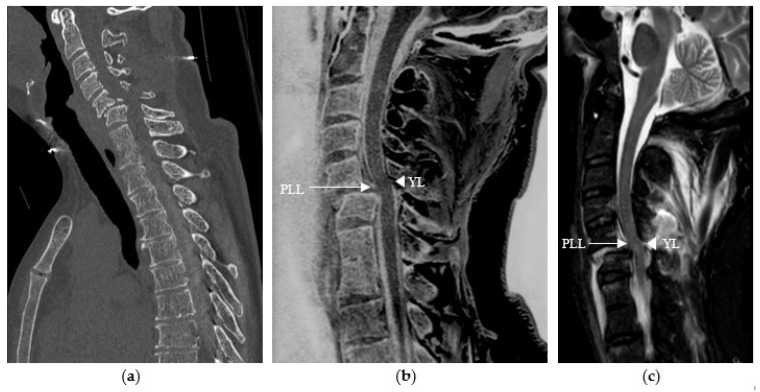
Traumatic cervical spine injury in a 58-year-old patient at C5–C6, (**a**) initially evaluated with sagittal CT reconstruction in which ligamentous damage could not be assessed. (**b**) The sagittal CT-like 3D mFFE sequence revealed damage to the posterior longitudinal ligament (PLL) and the yellow ligament (YL), visualized as a discontinuous white line (arrow and arrowhead), (**c**) which was confirmed by the remaining MRI sequences, particularly the sagittal T2 STIR sequence.

**Figure 4 diagnostics-14-02434-f004:**
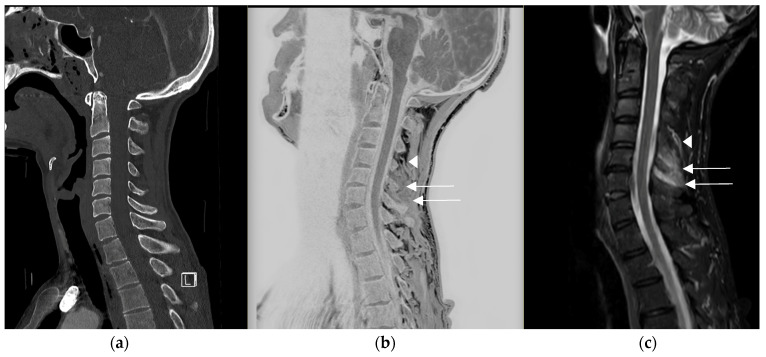
Traumatic cervical lesion in a 44-year-old patient at C6–C7, (**a**) initially evaluated with sagittal CT reconstruction for which, apart from the disc lesion, the analysis of the posterior ligaments was limited. (**b**) The CT-like sequence showed an edematous interspinous and supraspinous infiltration in C4–C5 and C5–C6 (see arrow), distinct from the vascular structures visible at the superior level in C3–C4 (arrowhead), (**c**) which was confirmed on the MRI protocol with the sagittal T2 STIR sequence.

**Table 1 diagnostics-14-02434-t001:** Three-dimensional mFFE sequence parameters.

	3D mFFE Cervical Spine	3D mFFE Thoracic or Lumbar Spine
Field of view	320 × 250 mm	320 × 250 mm
Acquisition voxel size	0.8 × 0.8 × 1.5 mm	0.8 × 0.8 × 1.5 mm
Reconstruction voxel size	0.42 × 0.42 × 0.75 mm	0.42 × 0.42 × 0.75 mm
Parallel imaging	CS-SENSE factor 2	CS-SENSE factor 2
Echo time (ms)	4.6 ms	4.6 ms
Repetition time (ms)	31 ms	35 ms
Delta echo time (ms)	5.8 ms	5.8 ms
Acquisition time	3 min 46 s	5 min 09 s

mm: millimetre; min: minute; ms: millisecond; s: second.

**Table 2 diagnostics-14-02434-t002:** Imaging parameters for vertebral fracture assessment.

Parameter	Description	Grading Scale
Genant classification	Semiquantitative visual grading of vertebral deformities	Grade 1 (20–25%) Grade 2 (25–40%) Grade 3 (>40%)
Anterior vertebral body height	Measured in the median sagittal plane, from the anterosuperior to the anteroinferior corner of the vertebral body, excluding osteophytes or fragments	mm
Posterior vertebral body height	Measured in the median sagittal plane, from the posterosuperior to the posteroinferior corner of the vertebral body, excluding osteophytes or fragments	mm
AO/Magerl classification	Classification of fractures in compression, distraction, and translation injuries according to Magerl et al. and Vaccaro et al.	A1: wedge compression A2: split A3 + 4: incomplete and complete burst B: distraction C: displacement or dislocation
Disc fracture	Fracture or traumatic displacement	Yes or no
Ligamentous injury		Yes or no
Diagnostic quality of the images	Likert scale	1: inadequate 2: poor 3: moderate 4: good 5: excellent

**Table 3 diagnostics-14-02434-t003:** Inter- and intra-observer agreement.

Parameters	Inter-Observer Agreement for CT Images	Inter-Observer Agreement for CT-like Images	Intra-Observer Agreement Between CT and CT-like Images (Observer 1)
AO classification (weighted kappa)	0.93321 95%CI (0.83866–1.0000)	0.91289 95%CI (0.81294–1.0000)	0.87027 95%CI (0.73079–1.0000)
Genant classification (weighted kappa)	0.94783 95%CI (0.84661–1.0000)	0.94643 95%CI (0.84161–1.0000)	0.79039 95%CI (0.60741–0.97338)
Anterior vertebral body height (ICC)	0.9684 95%CI (0.9284–0.9862)	0.9474 95%CI (0.8824–0.9769)	0.9600 95%CI (0.9098–0.9825)
Posterior vertebral body height (ICC)	0.9795 95%CI (0.9533–0.9911)	0.9692 95%CI (0.9302–0.9866)	0.9757 95%CI (0.9447–0.9894)

CI: confidence interval; ICC: intraclass correlation coefficient; CT: computed tomography.

## Data Availability

The original contributions presented in the study are included in the article, further inquiries can be directed to the corresponding author.

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
