# Peer review of "mFFE CT-like MRI Sequences for the Assessment of Vertebral Fractures"

_diagnostics, 2024, doi:10.3390/diagnostics14212434_

Round 1

Reviewer 1 Report

Comments and Suggestions for Authors

An interesting work on MRI sequences for the assessment of vertebral fracture. The topic is of clinical value and the manuscript contains a lot of valuable information; however, some aspects need to be clarified.

-Even if the results of your study are promising and 3D T1 multi echo FFE CT-like sequences could “substitute CT in the assessment of vertebral fractures”, nowadays it is not possible to replace CT in the setting of the traumatic patient. CT is faster than MR and allows rapid assessment of abdominal and thoracic organs and vessels beyond the bones, which is of primary importance in trauma setting. MR and CT are, therefore, complementary in the trauma. I suggest to rewrite this section.

-What is the total duration of the MR protocol including these sequences and what is the total duration of CT and MR examinations in these patients? Please add this information in the manuscript.

Comments on the Quality of English Language

There are some grammar mistakes, so moderate editing of English language is required.

Author Response

Dear Editor, dear Reviewers,

On behalf of all the authors, I would like to thank you for the time and effort you dedicated to reviewing our manuscript titled “mFFe CT-Like MRI Sequences for the Assessment of Vertebral Fractures”, which we had earlier submitted to Diagnostics. We are very grateful to both reviewers for their comments and queries, which were quite thoughtful and in-depth for us. Therefore, we have adapted the manuscript accordingly and incorporated the Reviewers’ suggestions, thereby improving the paper’s editorial structure and content, as requested. In addition, the manuscript has been entirely proofread by an English native speaker for improved clarity.

Please find below our point-by-point answers to the reviewer's queries and comments. In addition, we are enclosing the revised manuscript for your consideration.

We very much thank you for reconsidering our manuscript for publication and hope it will now find a definite place in Diagnostics.

Yours faithfully,

R1

Comment 1: Even if the results of your study are promising and 3D T1 multi echo FFE CT-like sequences could “substitute CT in the assessment of vertebral fractures”, nowadays it is not possible to replace CT in the setting of the traumatic patient. CT is faster than MR and allows rapid assessment of abdominal and thoracic organs and vessels beyond the bones, which is of primary importance in trauma setting. MR and CT are, therefore, complementary in the trauma. I suggest to rewrite this section.

Response 1: Thank you for your valuable comment. We have changed the paragraph in red between lines 205 and 213 (page 6) to include this nuance. We have also simplified the conclusion by avoiding stating that the CT-like sequence replaces CT, particularly for polytrauma patients who may have extra-osseous lesions that would require whole-body injected CT.

Comment 2: What is the total duration of the MR protocol including these sequences and what is the total duration of CT and MR examinations in these patients? Please add this information in the manuscript.

Response 2: Thank you for your comment. As every trauma case is different, we have given the example of the acquisition time for a patient in whom we performed a polytrauma whole-body protocol with late acquisition time and a complementary single-stack MRI to obtain an estimate of the shortest acquisition time when both examinations are combined.
These changes appear in red in the same paragraph of the manuscript between lines 212 and 213 (page 6).

Reviewer 2 Report

Comments and Suggestions for Authors

More information about mFFe CT Like MRI should be provided in the introduction section

In the material method section, patients excluded, the reasons for exclusion should be shown in a flow chart with their numbers and details

Some reference names related to interobserver agreement leveling are mentioned, but no formal source is given. This section should be edited, a formal source should be given and added to the references

This section should be reorganized by removing words that are the same as the title and those that are not directly related to the main subject of the study from the keywords section

The discussion section is too superficial. The first paragraph should include a summary of the study's purpose and most important findings. The following paragraphs should emphasize why this technique is needed compared to CT. Other studies conducted with this technique should be mentioned briefly. The conclusion section should be written as a single paragraph, emphasizing the most important findings. The bibliography subheading is meaningless and should be deleted. Limitations should not be given under the conclusion, but as the last paragraph of the discussion before the conclusion.

Author Response

Dear Editor, dear Reviewers,

On behalf of all the authors, I would like to thank you for the time and effort you dedicated to reviewing our manuscript titled “mFFe CT-Like MRI Sequences for the Assessment of Vertebral Fractures”, which we had earlier submitted to Diagnostics. We are very grateful to both reviewers for their comments and queries, which were quite thoughtful and in-depth for us. Therefore, we have adapted the manuscript accordingly and incorporated the Reviewers’ suggestions, thereby improving the paper’s editorial structure and content, as requested. In addition, the manuscript has been entirely proofread by an English native speaker for improved clarity.

Please find below our point-by-point answers to the reviewer's queries and comments. In addition, we are enclosing the revised manuscript for your consideration.

We very much thank you for reconsidering our manuscript for publication and hope it will now find a definite place in Diagnostics.

Yours faithfully,

R2

Comment 1: More information about mFFe CT Like MRI should be provided in the introduction section

Response 1: Thank you for your comment. The paragraph highlighted in green between lines 60 and 64 (page 2) provides further clarification.

Comment 2: In the material method section, patients excluded, the reasons for exclusion should be shown in a flow chart with their numbers and details

Response 2: A flowchart has been added as requested, referred to as Figure 1 in the manuscript.

Comment 3: Some reference names related to interobserver agreement leveling are mentioned, but no formal source is given. This section should be edited, a formal source should be given and added to the references.

Response 3: Thank you for your comment. The omission has been corrected and the appropriate reference (Landis and Koch, number 13) has been added in green (line 142, page 4).

Comment 4: This section should be reorganized by removing words that are the same as the title and those that are not directly related to the main subject of the study from the keywords section

Response 4: We have modified the ‘Keywords’ section to include other important keywords in green (lines 39-40). However, we think it is important to keep the word ‘CT-like’ for correct referencing of the article.

Comment 5: The discussion section is too superficial. The first paragraph should include a summary ofthe study's purpose and most important findings. The following paragraphs should emphasize why thistechnique is needed compared to CT. Other studies conducted with this technique should be mentioned briefly. The conclusion section should be written as a single paragraph, emphasizing the most important findings. The bibliography subheading is meaningless and should be deleted. Limitations should not be given under the conclusion, but as the last paragraph of the discussion before the conclusion.

Response 5: Thank you for your comment. We believe that there was a formatting error that affected the readability of this section. In fact, the discussion continued after subtitle 5, including paragraphs on disc lesions, reclassification of fractures, diagnostic quality of images, limitations, spectral imaging, and explanation of VNCa reconstruction. Only the last paragraph actually corresponds to the conclusion. We have corrected the formatting and removed the misplaced word ‘bibliography’.

We have also modified the beginning of the discussion to include a more precise comparison with the literature (lines 190-202, page 6).

Besides, we have simplified the conclusion by avoiding stating that the CT-like sequence replaces CT, particularly for polytrauma patients who may have extra-osseous lesions that would require whole-body injected CT.

Round 2

Reviewer 1 Report

Comments and Suggestions for Authors

No further comments.

Reviewer 2 Report

Comments and Suggestions for Authors

The article has improved after revisions.